# Comparison of the Effect of Different Resistance Training Frequencies on Phase Angle and Handgrip Strength in Obese Women: A Randomized Controlled Trial

**DOI:** 10.3390/ijerph17041163

**Published:** 2020-02-12

**Authors:** Stefania Toselli, Georgian Badicu, Laura Bragonzoni, Federico Spiga, Paolo Mazzuca, Francesco Campa

**Affiliations:** 1Department of Biomedical and Neuromotor Sciences, University of Bologna, 40126 Bologna, Italy; stefania.toselli@unibo.it (S.T.); laura.bragonzoni4@unibo.it (L.B.); Federico2907@gmail.com (F.S.); francesco.campa3@unibo.it (F.C.); 2Department of Physical Education and Special Motricity, University Transilvania of Brasov, 500068 Brasov, Romania; 3Unit of Internal Medicine, Diabetes and Metabolic Disease Center, Romagna Health District, 47921 Rimini, Italy; Paolo.mazzuca4@unibo.it; 4Department for Life Quality Studies, University of Bologna, 47921 Rimini, Italy

**Keywords:** bioimpedance, BIVA, body composition, R-Xc graph

## Abstract

Phase angle (PA) is a strong predictor of sarcopenia, fragility, and risk of mortality in obese people, while an optimal muscular function and handgrip strength (HS) are required to perform different daily activities. Although there is a general agreement that resistance training improves health status in obese people, the optimal weekly training frequency for PA and physical performance parameters is not clear. This study aimed to compare the effects of different weekly resistance training frequencies performed over a 24 week exercise program on PA and HS in obese people. Forty-two women (56.2 ± 9.1 years, body mass index (BMI) 37.1 ± 4.9 kg/m^2^) were randomly allocated to one of two groups: a group with a high weekly training frequency of three times a week (HIGH, n = 21) and a group that performed only one weekly session (LOW, n = 21). The groups trained with an identical exercise intensity and volume per session for 6 months. Before and after the intervention period, the participants were assessed for anthropometric measures, bioimpedance analysis, and HS. There was a significant group × time interaction (*p* < 0.05) for waist circumference, bioimpedance reactance divided by body height (Xc/H), PA, and HS measures. In addition, only the HIGH group increased Xc/H, PA, and HS after the intervention period (*p* < 0.05), even after adjusting for weight loss and menopausal status. Physical exercise performed three times a week promotes better adaptations in PA and HS when compared with the same program performed once a week in obese women.

## 1. Introduction

In recent years, obesity has become widely recognized as a growing public issue requiring urgent action, especially in high-risk groups. Currently, more than 1.5 million people are obese or overweight, reaching about 25% of the population in Canada and 35% in the United States [1,2]. Moreover, in most countries of northern Europe, including the UK and Scandinavian countries, as well as southern European countries, the rate of obesity has more than doubled over the past 30 years [3,4,5]. Obesity is associated with increased morbidity, and disability, from cardiovascular disease, diabetes, cancer, hypertension, osteoarthritis, and musculoskeletal disorders [6]. Although bariatric surgery remains the most effective treatment to reduce and maintain weight loss, as well as improve comorbidities and mortality, physical activity is recommended as the first step to achieving weight loss and reducing obesity-related comorbidities in subjects with severe obesity [7,8,9]. 

The bioimpedance vector analysis (BIVA) is used in clinical and sports fields to study the change in body fluids and nutritional state [10,11,12,13]. It represents the raw bioimpendance parameters (resistance (R) and reactance (Xc)) as a point on the R–Xcgraph in which both length and slope are considered. The vector slope represented by the bioelectrical phase angle (PA) is an indicator of cell membrane integrity and extracellular/intracellular(ECW/ICW) ratio [14,15]. Recent studies on obese people have taken into consideration PA and its relationship to different health status measurements, suggesting it as a biomarker to quantify inflammation, which might help in identifying high-risk patients [16,17]. In particular, it has been shown that obese women with a low PA tertile have high fat mass with high levels of glucose and higher cardiovascular risk factors [18]. On the other hand, in relation to muscle function, improvements in PA are also associated with increases in strength, in particular that of hand grip (HS) [19,20]. In this regard, HS and muscle strength, indicators of muscle quality, have shown to be more significant than muscle mass in estimating mortality risk [21].

Previous studies have shown that the PA can be modulated by exercise. These studies, which have offered resistance training as physical activity, have been used for training protocols with a frequency of two or three times a week [22,23,24,25]. The World Health Organization (WHO) recommends that all adults should engage in a minimum of 150 to 300 minutes of moderate-intensity physical activity or at least 75 minutes of vigorous-intensity exercise throughout the week, involving the major muscle groups [26].

To the best of our knowledge, no study compared the effects of different training frequencies on BIVA patterns and muscular function in obese people. Therefore, the aim of this study was to evaluate the effects of a resistance training program of one or three times a week on PA and HS in obese women. 

## 2. Methods 

### 2.1. Experimental Approach to the Problem

A randomized controlled trial was carried out over a period of 28 weeks. The first 2 weeks and the last 2 weeks were used for measurements and evaluation. The participants were randomly divided into one of two groups: a group with a high weekly frequency of three times a week (HIGH) and a group where participants attended the program with a low weekly rate of one workout per week (LOW). A blinded researcher was responsible for generating random numbers (random.org) for group placement. All measures were taken in the hospital department, while the training program was performed in a contracted sports center (from January to June 2017). Our study was designed, implemented, and reported per the CONSORT statement [27]. This study was registered at www.clinicaltrials.gov (registration code: NCT03410329).

### 2.2. Participants

An a priori power analysis was conducted to determine the sample size for the study (G*Power 3.1.9.2, Germany). The primary outcomes investigated in this research were phase angle, flexibility, and handgrip strength. The following design specifications were considered: α = 0.05; (1-β) = 0.8; effect size f = 0.25; test family = F test and statistical test = ANOVA repeated measures. The sample size estimated according to these specifications was 34 subjects, 17 in each group. Thus, we selected 60 women patients referred to the Lifestyle program from the Department of Endocrinology of the “Infermi” hospital of Rimini, who volunteered to participate in this study. The inclusion criteria for participation in the treatment program were as follows: body mass index (BMI) greater than or equal to 30. We did not accept candidates who smoked, had cardiovascular disease or any other major illness, or were taking medications that could have affected the results.

All participants gave written informed consent after a detailed description of the study procedures was provided. The Project was conducted in accordance with the guidelines of the declaration of Helsinki and was approved by the local Bioethics Committee (approval code: 12012016).

### 2.3. Procedures

The supervised exercise program was specifically devised to favor a stepwise incremental approach to exercise. After enrollment, patients exercised in a gymnasium where a trained team supervised the individual progressive exercise program (120 min, for 24 weeks). At the beginning of each session, a series of respiratory, proprioceptive, and flexibility exercises were carried out, followed by progressively increasing resistance training. Resistance training consisted of an isotonic machine circuit for major muscle groups (four sets of 8–12 repetitions at 10RM for leg press, leg extension, lat machine, low row, chest press, pectoral machine, and shoulder press, with each set completed in approximately 30sec with 1min rest), with the 10RM level determined during the initial session by the exercise physiologist. Starting workload levels for each piece of equipment were tested by participants and if more than 10 repetitions were achieved, the weight was increased and after a short rest, the participants tried again. Likewise, if less than 8 repetitions were achieved, the weight was decreased and after a short rest, the participants tried again. The last part of each session was devoted to stretching. 

The anthropometric traits were weight, height, and waist circumference (WC). All anthropometric measurements were taken according to standard methods [28]. Height was recorded to the nearest 0.1 cm with a stadiometer, and weight was measured to the nearest 0.1 kg with a high-precision mechanical scale. BMI was calculated as the ratio of body weight to height squared (kg/m^2^). WC was taken to the nearest 0.1 cm with a tape measure. 

Bioelectrical impedance was measured with a phase-sensitive impedance plethysmograph (BIA-101 Anniversary Sport Edition, Akern-RJL Systems, Florence, Italy). The device was used to obtain whole-body R and Xc at a single frequency (50 kHz) and was calibrated every morning, using a calibration circuit procedure of known impedance (R = 380 Ohm, Xc = 47 Ohm, 1% error) supplied by the manufacturer. Standard whole-body tetrapolar measurements were taken according to conventional procedures established in the literature [29]. Participants were instructed to urinate about 30 min before the measures, refrain from ingesting food or drink in the last 4 hours, avoid strenuous physical exercise for at least 24 hours, and refrain from consumption of alcoholic and caffeinated beverages for at least 48 hours. PA was calculated as the arctangent of Xc/R × 180°/π. Bioimpedance vector analysis was carried out using the BIVA method, normalizing R and Xc parameters for height (H) in meters [30]. Fat mass percentage (F%) was predicted using the software Bodygram® (AkernSrl., Pontassieve, Florence, Italy).

Left and right handgrip strengths were measured to the nearest 0.5 kg with a mechanical dynamometer (Takei K.K. 5001, Takei Scientific Instruments, Ltd., Niigata City, Japan) in a sitting position by holding the dynamometer at a 90degree flexion of their elbow. Maximal readings of three measurements from both hands were recorded. Dominant handgrip strength (DHS) and total handgrip strength (THS) were measured. THS was as summed from readings of both hands [31].

### 2.4. Statistical Analyses

Normality was verified with the Shapiro–Wilk test. The main hypothesis was interpreted using analysis of variance for repeated measures (ANOVA two-way) for between- and within-group comparisons. When F-ratio was significant, Bonferroni’s post hoc test was used for the identification of specific differences in the variables. Effect size (ES) was calculated as post-training mean minus pre-training mean divided by the pooled standard deviation. The ES values were then classified as follows: 0.20–0.49 was considered small, 0.50–0.79 was considered moderate, and ≥ 0.80 was considered large. Two-way analysis of covariance (ANCOVA) for repeated measures was applied for comparisons, using weight loss and menopausal status as covariates. The paired, one-sample Hotelling’s T^2^test was performed to determine if the changes in the mean group vectors (measured at the first and second time points) were significantly different from zero (null vector). A 95% confidence ellipse excluding the null vector indicated a significant vector displacement. Statistical significance for all analyses was defined as *p* < 0.05. SPSS (SPSS 23.0.0.0; SPSS Inc., Chicago, IL, USA) was used for all statistical calculations.

## 3. Results

The characteristics of the participants at baseline are shown in Table 1. The flow chart with a schematic representation of participant allocation is presented in Figure 1. 

Twenty-three women had been randomly assigned to a HIGH group (age 53.7 ± 9.3 years, BMI 37.9 ± 4.1 kg/m^2^) and 23 to the LOW group (age 58.7 ± 8.5 years, BMI 36.2 ± 5.7 kg/m^2^).

No significant differences in age (t = −1.540, *p* = 0.135) and BMI (t = 0.977, *p* = 0.337) were found between the two groups before the intervention period.

There was a significant group by time interaction for WC, Xc/H, PA, DHS, and THS. Post hoc revealed that in both groups, body weight, WC, and F% significantly decreased from before to after the intervention period, whereas Xc/H, PA, DHS, and THS increased only in the HIGH group after 24 weeks (Table 2). After adjusting for weight loss and menopausal status, as covariates, the group by time interaction for PA, DHS, and THS remained significant (*p* < 0.05).

In Figure 2 is illustrated the bioelectrical impedance mean vector displacements against the 95%, 75%, and 50% tolerance ellipses of the reference population [32]. 

The paired one-sample Hotelling’s T^2^ test indicated a significant difference in the mean vectors between the first measurement and the second measurement only for the HIGH group (HIGH, T^2^ = 35.9, F = 16.7, *p* = < 0.01, Mahalanobis D = 1.5; LOW, T^2^= 1.7, F = 0.8, p = 0.5, Mahalanobis D = 0.3) (Figure 3).

## 4. Discussion

The aim of this study was to compare the effects of different weekly resistance training frequencies performed over a 24 week exercise program on PA and HS in obese people. Although both groups had an improvement trend after 26 weeks of intervention, the major effects on the examined parameters were measured in the group that performed the training program at a higher weekly frequency. In fact, although body weight, WC, and F% significantly decreased in both group after the intervention period, Xc/H, PA, DHS, and THS increased only in the group that performed the training program at a higher weekly frequency.

In the physical activity prescription, this study was based on the upper limit of the amount recommended by the American College of Sports Medicine in terms of both frequency and volume per session [33] and on the minimum frequency of physical activity suggested by WHO [26].Our resistance training program has led to weight loss and a reduction in fat mass in participants, and this is in line with the findings of other authors [34,35]. However, there are conflicting reports in the literature on whether or not resistance training induces fat mass loss. Some studies report a statistically insignificant trend or no change in fat mass after a resistance training program [22,36]. These discrepancies in the results may be due to experimental designs that did not include a calorie restriction program or the fact that they considered fat mass in absolute and non-percentage terms with respect to body weight. 

The literature is abundant about resistance training studies, but very little is known regarding the effects of different frequencies of weekly training on PA and HS. As demonstrated by our results, the choice of weekly training frequency causes different effects on the examined variables. However, it is well known that aging and a sedentary lifestyle cause a decrease in PA and HS [19,37], therefore even the results obtained by the LOW group, where no significant parameter changes were shown, are still noteworthy. Ribeiro et al. [37] showed improvements in PA following resistance training exercises carried out 3 times a week in obese women. In their study, the results after 8 weeks of training showed a significant reduction in resistance and no change in reactance. In our study, after 24 weeks of resistance training, even if R tended to decrease in the HIGH group, this change did not reach statistical significance. On the other hand, our results showed a significant increase in Xc after the intervention period. Changes in PA are determined by alteration in cellular membrane integrity (Xc) or body fluid (R) or a combination of both. Theoretically, high Xc values are expected from a BIA measurement in healthy membranes with higher integrity. The healthy cell membranes act as capacitors by storing the current and releasing it. In fact, Nescolarde et al. [38] showed that Xc and PA decrease after a muscle injury. Thus, Xc is proportional to cellmembrane integrity, with Xc and PA decrements occurring when the cell membrane is compromised. Our hypothesis states that in our study, the increase in PA was caused by an improvement in cellular membrane integrity, causing increases in Xc and a reduction in the ECW/ICW relationship, which is inversely proportionate to PA [14,15,39]. Furthermore, since R is inversely proportional to body fluid content, it could decrease absolutely following a reduction in body weight, causing a tendency to increase R and to elongate the vector in the R-Xc graph (Figure 2). On the contrary, increase in PA and therefore in the ICW/ECW ratio could cause a reduction in R, resulting in no significant variable change. However, as in our case, nutritional habits during the intervention period have not been monitored by Ribeiro et al. [37] and this may have caused a disparity of changes in Xc and R, which in both cases led to an increase in AP after a resistance training program in obese women. Recently, Norman et al. [40] have shown significant correlations between PA with muscle strength and other physical performance parameters, including the capability to perform daily activities. HS decreases with inactivity and aging [19], therefore the tendency to increase DHS and THS measured in the LOW group can be a noteworthy result. Our view is that high-frequency training is crucial in the development of strength, which plays also an important role in cardiovascular health. On this point, Lee et al. [41] reported that HS is correlated with cardiometabolic risk factors including blood pressure, triglyceride, HDL cholesterol, HbA1c, and fasting glucose.

Despite the encouraging results obtained in this study, some limitations are present and should be considered. First, our results are applicable to BIA equipment using the 50 kHz frequency and to a similar population. In fact, in a recent study by Silva and colleagues [42], a comparison between single (50 kHz) and multi-frequency devices was conducted in a highly active population. The authors found that multi-frequency instruments provided significantly lower values of R and Xc but higher values of PA at 50kHz.Second, although BIA is used in clinical and sports fields to reflect body fluids, nutritional state, and phase angle [29,43,44,45], it is not a direct measurement method of body composition. In particular, in this study, F% was determined from the bioimpedance raw parameters rather than by a gold standard method. Third, glucose level and cardiovascular risk factors changes were not monitored during the experiment. Lastly, we were not able to monitor physical activity levels outside of the study environment and to track dietary intake throughout the study, though individuals were asked to maintain their usual lifestyle habits. Further evidence on the subject could be collected, contributing to this particular field of research and supporting the work of clinicians working with physical exercise for obese people.

## 5. Conclusions

Resistance training is effective to induce improvement in body composition, but increases of PA and HS can only be achieved with a high training frequency. From our observations, a high training frequency gave greater benefits for the health status than a training program with only one weekly session. Moreover, BIVA can help to evaluate and monitor health status and effectiveness of physical activity programs. There is considerable scope for researchers and trainers to contribute to reducing the problem of obesity and its related consequences. 

## Figures and Tables

**Figure 1 ijerph-17-01163-f001:**
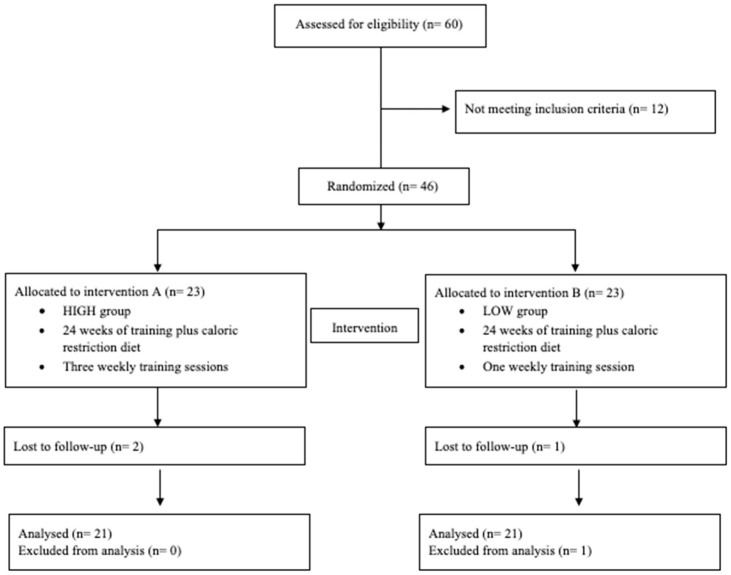
Flow chart.

**Figure 2 ijerph-17-01163-f002:**
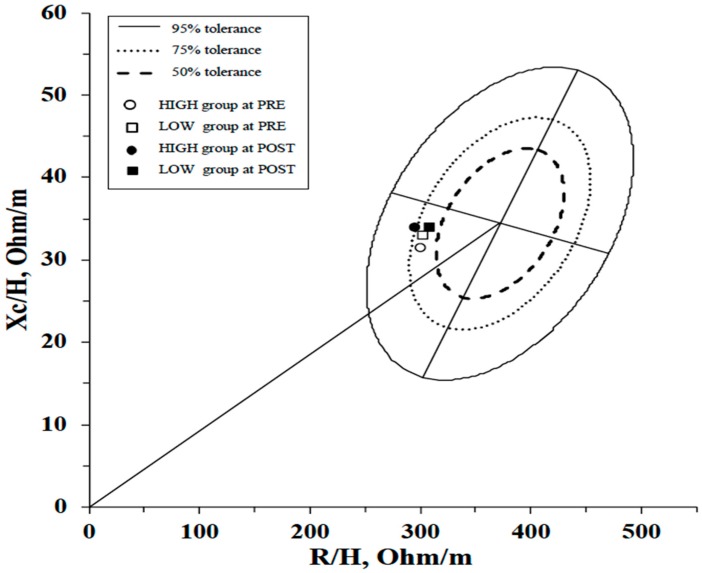
R-Xc graph and vector displacements from before (PRE) to after (POST) the intervention period using the mean R/H and the mean Xc/H plotted on the reference population tolerance ellipses [32]. HIGH: the group with a high-weekly training frequency of three times a week; LOW: the group that performed only one weekly session.

**Figure 3 ijerph-17-01163-f003:**
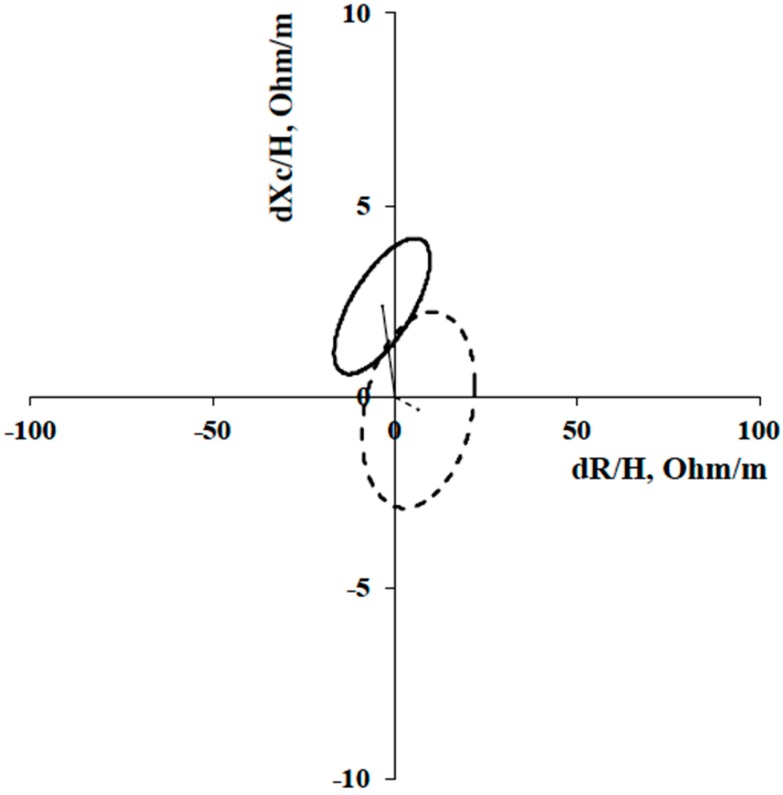
Paired Graph and impedance vector displacements with 95% confidence ellipses from before to after the intervention period. Continuous and dotted lines for the HIGH group and the LOW group, respectively. HIGH: the group with a high-weekly training frequency of three times a week; LOW: the group that performed only one weekly session.

**Table 1 ijerph-17-01163-t001:** General characteristics of participants before the intervention period.

Variable	HIGH	LOW
Age (years)	53.7 ± 9.3	58.7 ± 8.5
BMI (kg/m^2^)	37.9 ± 4.1	36.2 ± 5.7

Note: Data are expressed as mean and standard deviation. BMI: body mass index.

**Table 2 ijerph-17-01163-t002:** Two-way ANOVA for the comparison between the groups before and after the intervention.

Variable	HIGH (n=21)		LOW (n= 21)		ES^ §^	Interaction *P*-Value	SP
	Before	After	Before	After			
Weight (kg)	96.8 ± 13.1	89.0 ± 12.4 *	88.4 ± 13.7	83.4 ± 10.8 *	−0.57	0.12	0.33
WC (cm)	108.4 ± 12.3	99.3 ± 11.5 *	107.9 ± 11.7	103.1 ± 11.3 *	−0.91	0.01	0.67
F (%)	39.7 ± 3.0	36.4 ± 3.3 *	39.3 ± 3.8	37.1 ± 3.9 *	−0.58	0.12	0.34
R/H (Ω)	299.1 ± 28.4	295.6 ± 28.8	303.3 ± 32.4	309.8 ± 33.8	−0.51	0.16	0.27
Xc/H (Ω)	31.3 ± 3.0	33.7 ± 2.0 *	33.9 ± 6.6	33.8 ± 6.23	0.77	0.04	0.52
PA (degrees)	6.0 ± 0.5	6.5 ± 0.5 *	6.2 ± 0.8	6.4 ± 0.6	0.86	0.02	0.62
DHS (kg)	24.0 ± 5.5	28.1 ± 5.4 *	23.0 ± 5.6	24.0 ± 5.2	1.03	<0.01	0.78
THS (kg)	45.9 ± 52.0	52.0 ± 9.4 *	43.5 ± 10.3	45.6 ± 10.7	0.80	0.03	0.56

Note: Data are expressed as mean and standard deviation. * *P *< 0.05 vs. before, WC: waist circumference, F: fat mass, R/H: resistance divided by body height, Xc/H: reactance divided by body height, PA: phase angle, DHS: dominant handgrip strength, THS: total handgrip strength, SP: statistical power, §: the Hedges’ g effect size was used.

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
