# Peer review of "Comparison of the Effect of Different Resistance Training Frequencies on Phase Angle and Handgrip Strength in Obese Women: A Randomized Controlled Trial"

_ijerph, 2020, doi:10.3390/ijerph17041163_

Round 1
Reviewer 1 Report
Based on the hypothesis that the increase in PA were caused by an improvement in cellular membrane integrity, causing increases in Xc and a reduction in the ECW/ICW relationship, the manuscript measured the effects of HIGH and LOW resistance training frequencies on PA and HS in obese ladies. The authors found that HIGH training group promoted better adaptations in PA and HS when compared with LOW group in obese women.
The study was well designed and the manuscript was well written. Before making a decision, however, there were some issues to be clarified first.
Page 2 Line 46
Please provide the full name of BIVA.
Page 3 Line 130-132
Dominant handgrip strength (DHS) and absolute handgrip strength (AHS) were measured. AHS was as summed from readings of both hands [31].
The meaning of “A” from AHS was strange. Even the article of Choquette et al was cited, there was no any word to describe AHS in the text. I know the AHS is summation of both handgrip strengths, but I still wonder why not use “total or both” to replace “absolute.” In my opinion, either THS or BHS is better than AHS.
Fig 3. The figure did not tell me what HIGH group or LOW group is, and where the before and after is. Please describe them in figure legend clearer.
Page 4 Line 158-160
The results showed Xc/H, PA, DHS, and AHS increased only in the HIGH group after 24 weeks (Table 1). After adjusting for weight loss and menopausal status, as covariates, the group by time interaction for PA, DHS, and AHS remained significant (p<0.05).
Sadly, there is no any sentence in DISCUSSION to describe why the HIGH resistance training enhanced the Xc/H and PA, rather than R/H. In addition, I am wondering why the HIGH resistance training enhanced the DHS and AHS, but LOW group did not. Please offer sentences for explanation.
Page 7 Line 204
PA are determined “byalteration” in …
Change to -- > PA are determined “by alteration” in …
Author Response
Dear Reviewer,
I attach the response point-by-point.
Thank you!

Reviewer 2 Report
This is an interesting article about proper frequency of physical exercise to improve phase angle and handgrip strength in obese women. However, there are some questions need authors to clarify.
Methods section
1. Handgrip strength is affected by comorbidity such as osteoarthritis or rheumatoid arthritis. Have you excluded subjects with these disease? If not, explain how to adjust the impact of such comorbidities.
2. Phase angle generally shows higher value in older people and male. Have you adjusted age and sex in the results?
3. Please describe the details of inclusion and exclusion criteria.
4. Please explain how you controlled factors that could affect phase angle and handgrip strength, such as nutrition, calorie intake, and physical activity.
5. In introduction section, author mentioned that PA is related with high glucose level, cardiovascular risk factors, and inflammation. Did you consider these metabolic aspects in trials?
Results section
6. In table 1, even though BMI of High group and Low group was not significantly different, weight was much higher in HIGH group. As weight increases, BIA tends to change better with exercise. How could you overcome this limitation?
7. It would be better to separate Table 1. Please show baseline characteristics in table 1 and show the results in table 2.
8. The author mentioned that PA is associated with muscle strength. Was there any relationship between PA variance and handgrip strength variance?
Discussion section
9. Please extend the discussion section with mechanism how resistance exercise improves cell membrane integrity.
10. Although bioimpedence analysis is easy and good method to measure body composition, it has several limitations. Please mention the limitation of BIA in discussion section.
Author Response

(The authors gave the same response as above.)

Round 2
Reviewer 1 Report
The manuscript investigated the effects of HIGH and LOW resistance training frequencies on PA and HS in obese ladies in order to verify the fact that improvement in cellular membrane integrity will increase PA, elevate Xc and reduce the ECW/ICW relationship. The authors found that HIGH training group promoted better adaptations in PA and HS when compared with LOW group in obese women.
The study was well designed and the revised manuscript was well written. All the queries have been point-to-point responded and acceptable. I recommend accepting this article.
A minimal error present in Table 2, as below. Please correct it. I do not need to check it again.
“THS: Tbsolute” - - > Change to “THS: total”
Author Response
The authors would like to thank to reviewers for their precious and constructive advice. We found the comments very helpful in improving the paper.
We have attached our responses from each point that has been addressed.

Reviewer 2 Report
We reviewed the revised version of manuscript, but there are still several points to be revised. 1. Please correct wrong spacing between words. For example, Page 6 in figure 2 footnote “HIGH: Thegroup” Page 8 Line 236 “with higherintegrity” Page 8 Line 238 “to cellmembrane” 2. Page 5 in table 2 footnote, “THS: Tbsolute” may be misspelled. Please correct it. 3. Page 8 Line 257 “First, our resultsare applicable to BIA equipment using the 50 kHz frequency and toa 257 similar population.” I don’t understand what you mean by this sentence. Although, BIA is used in clinical and sports fields to reflect body fluids, nutritional state, and phase angle, it is not direct measurement method of body composition or phase angle. Please mentioned that BIA has these limitations.Author Response
The authors would like to thanks the reviewers for their precious and constructive advice. We found the comments very helpful in improving the paper.
I attach the responses point-by-point.
